# Habitat Imaging of Tumors Enables High Confidence Sub-Regional Assessment of Response to Therapy

**DOI:** 10.3390/cancers14092159

**Published:** 2022-04-26

**Authors:** Paul David Tar, Neil A. Thacker, Muhammad Babur, Grazyna Lipowska-Bhalla, Susan Cheung, Ross A. Little, Kaye J. Williams, James P. B. O’Connor

**Affiliations:** 1Division of Cancer Sciences, University of Manchester, Manchester M13 9PT, UK; paul.tar@manchester.ac.uk (P.D.T.); grazyna.lipowska-bhalla@manchester.ac.uk (G.L.-B.); susan.cheung@manchester.ac.uk (S.C.); ross.little@manchester.ac.uk (R.A.L.); 2Division of Informatics, Imaging and Data Sciences, University of Manchester, Manchester M13 9PT, UK; neil.thacker@manchester.ac.uk; 3Manchester Pharmacy School, Division of Pharmacy and Optometry, University of Manchester, Manchester M13 9PT, UK; muhammed.babur@manchester.ac.uk (M.B.); kaye.williams@manchester.ac.uk (K.J.W.); 4Department of Radiology, The Christie Hospital NHS Trust, Manchester M20 4BX, UK; 5Division of Radiotherapy and Imaging, The Institute of Cancer Research, London SM2 5NG, UK

**Keywords:** cancer, imaging, modelling, statistics, machine learning, diffusion weighted MRI

## Abstract

**Simple Summary:**

Medical imaging techniques such as magnetic resonance imaging (MRI) are powerful tools that can map and measure tumor behavior in great detail. In particular, MRI can provide information about differences present within and between tumors that have a notionally similar type. At present, such imaging techniques are underused in assessment of cancer treatments, often because complicated spatial patterns present in each individual tumor mask individual responses to therapy. In this study we use mathematical modeling to assess tumors derived from 5 different mouse models of cancer. The modeling technique detected response to therapy in individual tumors and for different types of drug and radiation therapy, which was not possible using standard analysis of MRI data, where only group effects are detectable. Our results have potential to reduce the use of animals in medical research. They also enable a new high throughput MRI-based analysis of tumor models undergoing evaluation with new therapies.

**Abstract:**

Imaging biomarkers are used in therapy development to identify and quantify therapeutic response. In oncology, use of MRI, PET and other imaging methods can be complicated by spatially complex and heterogeneous tumor micro-environments, non-Gaussian data and small sample sizes. Linear Poisson Modelling (LPM) enables analysis of complex data that is quantitative and can operate in small data domains. We performed experiments in 5 mouse models to evaluate the ability of LPM to identify responding tumor habitats across a range of radiation and targeted drug therapies. We tested if LPM could identify differential biological response rates. We calculated the theoretical sample size constraints for applying LPM to new data. We then performed a co-clinical trial using small data to test if LPM could detect multiple therapeutics with both improved power and reduced animal numbers compared to conventional *t*-test approaches. Our data showed that LPM greatly increased the amount of information extracted from diffusion-weighted imaging, compared to cohort *t*-tests. LPM distinguished biological response rates between Calu6 tumors treated with 3 different therapies and between Calu6 tumors and 4 other xenograft models treated with radiotherapy. A simulated co-clinical trial using real data detected high precision per-tumor treatment effects in as few as 3 mice per cohort, with *p*-values as low as 1 in 10,000. These findings provide a route to simultaneously improve the information derived from preclinical imaging while reducing and refining the use of animals in cancer research.

## 1. Introduction

Medical imaging can provide serial whole volume assessment of tumors by producing spatially resolved maps of sub-units termed ‘voxels’ [1]. Tumors exhibit spatial variation in genetics leading to varying proteomic and metabolomic expression. This results in multiple microenvironments, which have been termed ‘tumor habitats’ [2], with habitat-specific voxel data distributions.

Imaging is frequently used in drug and radiotherapy evaluations [3]. Common modalities (MRI, PET, CT) have numerous different techniques yielding many biomarkers [4]. Usually these data are acquired on a voxel-wise basis, covering part or all of a lesion [5]. Data are often summarised as a mean or median, or distribution parameter such as standard deviation (SD) or percentile. ANOVA or *t*-tests then assess differences between control and treated parameters at a cohort-level. When effects are subtle it is difficult to identify changes in individual tumors.

Tumor heterogeneity creates problems for this image analysis approach. Simple parameter summaries of the ADC distribution, such as mean value or standard deviation, discard information regarding non-Gaussian behavour, such that tumors with equal volume, mean and 95th percentile can have very different histograms (Figure 1A). Parameters are affected by multiple co-existant habitats, each contributing in different ways. Treated tumors contain components that have changed following therapy and those that have changed due to natural history growth alone (Figure 1B). This effect is seen, but hidden in real visual data (Figure 1C) where differences in imaging biomarkers can be mapped in both control and radiotherapy (RT) treated tumors.

Various methods have been described that segment tumors into habitats [1]. Some approaches acquire several types of data (e.g., DWI, DCE-MRI and native T1 or T2) and use data driven methods such as PCA and clustering to define voxels with similar characteristics to one another [6,7]. Alternative approaches impose a priori cut points on image data [8,9,10]. These methods typically operate in a spatial domain and must contend with issues of non-uniqueness of solutions and the ambiguity of habitat boundaries.

Previously, we presented Linear Poisson Modelling (LPM) to analyse histograms in physical [11] and biological [12,13] sciences. In this latter study, LPM identified responding volumes of tumors treated with RT in xenograft models of colorectal cancer. The technique provided substantial additional power over *t*-tests. Our approach uses distribution variations to estimate a lower-bound of volume changes associated with treatment on a per tumor basis, as opposed to cohort level only summaries provided by *t*-tests and ANOVA [13]. LPM describes admixtures of histograms and can build models from ‘small data’ datasets of around 20 examples. This is useful in preclinical trials to reduce cost and to comply with the Reduction, Refinement and Replacement of animals in medical research (3Rs) [14].

Here, we investigate the limits of LPM for pre-clinical data analysis. We show how LPM applied to medical imaging can reduce the numbers of animals needed in studies while achieving very high statistical significances when detecting tumor changes. Through the use of new power calculations and simulations, we present evidence that experimental study designs with very small numbers of mice can be feasible. We then confirm practical application on real data by simulating a prospective co-clinical trial [15] using real data.

## 2. Materials and Methods

Using *t*-testing and LPM, we analysed preclincial murine xenograft models of human cancer and also simulated data. In each experiment we derived the MRI biomarker apparent diffusion co-efficient (ADC) [16]. Studies were performed in compliance with NCRI Guidelines for the welfare and use of animals in cancer research [14] and with Licences issued under the UK Animals (Scientific Procedures) Act 1986 (PPL 40/3212) following local Ethical Committee review.

Tumors belong to cohorts that were either control or treated. LPM identified volumes within ADC distributions that were unaffected or affected by treatment, by determining if treatment distributions differ from control distributions. These data can be used to define biological response, although this term does not imply subsequent clinical benefit. The affected volume is only a lower-bound, as it is possible that further biological changes are happening, but are not evident from the ADC distribution or the changes in ADC are not discernably different from controls.

### 2.1. Preparation of Tumor Cohorts

The majority of experiments were performed in the non-small cell lung cancer model Calu6. Details of the tumor propagation are provided in Section A.1. When tumors reached 200–300 mm${}^{3}$ in size by calliper measurement, mice entered the study and were randomized to different cohorts.

Initially, we analysed four groups: (a) Sham radiotherapy and saline control (*N* = 15); (b) Treatment with tumor-localised radiotherapy (RT) in a single 10 Gy fraction (*N* = 9); (c) Treatment with fractionated RT in five daily 2 Gy, with concurrent cisplatin on the first day of RT (fractionated chemo-radiotherapy, abbreviated as FCRT; *N* = 6); and (d) Treatment with 50 mg/kg/day of the oxygen consumption modifying agent atovaquone 3 with 2% DMSO and 0.1% carboxymethyl-cellulose in drinking water (*N* = 14). Scans were performed on days 0, 3 and 7 (visit 1 [V1], visit 2 [V2] and visit 3 [V3]).

We also performed a subsequent prospective co-clinical trial in four further groups: (e) Saline control: mice with similar growth characteristics to the previous control mice (*N* = 3; termed ‘strongly representative controls’) and mice with faster growth or larger entry size (>350 mm${}^{3}$) than the previous control mice (*N* = 4; termed ‘less representative controls’); (f) Treatment with a combination of atovaquone (ATV) 50 mg/kg/day followed by fractionated RT (*N* = 3); (g) Treatment with one dose of 60 mg/kg intraperitoneal injection of the hypoxia activated prodrug AQ4N (banoxantrone) 4 in saline (*N* = 3); (h) Treatment with two doses of 30 mg/kg of banoxantrone (*N* = 3). Here, mice were scanned at days 0 and 3 (visit 1 [V1], visit 2 [V2]).

In addition, we evaluated the effect of single 10 Gy fraction of RT compared to sham control in four further xenograft tumor models. These were two further ATCC human xenograft cell lines (the brain tumor U87 model and the colorectal cancer HCT116 models) as well as two syngeneic cell lines (the colorectal cancer CT26 model and the breast cancer 4T1 model). Further details of these models can be found in Section A.1.

For mice undergoing RT, treatment was administered using a metal-ceramic MXR-320/36 X-ray machine (320 kV, Comet AG, Switzerland). Irradiation was delivered at a dose rate of 0.75 Gy/min. Mice were turned around halfway through the procedure to ensure a uniform tumor dose.

### 2.2. MRI Acquisition and Analysis

MRI was performed on a 7T Magnex instrument interfaced to a Bruker Avance III console and gradient system, using a volume transceiver coil. Following localisation with a $T_{2}$-weighted anatomical sequence, we performed diffusion-weighted imaging (TR/TE = 2250/20 ms; α= 90${}^{\circ }$; b values 150, 500 and 1000 s/mm${}^{2}$ along one diffusion direction; matrix 64 × 64. The FOV, in-plane resolution, number of slices and slice thickness for each xenograft model are shown in Table 1 along with the resultant voxel volumes. We calculated voxel-wise values of ADC across the tumor using least squares fitting on the equation $S=S_{0}e^{-bD}$, where $S_{0}$ represents the signal intensity in the absence of a diffusion sensitising gradient, *S* the signal intensity for a particular *b* value, *b* the numerical value in s/mm${}^{2}$ and *D* the apparent diffusion coefficient (mm${}^{2}$/s).

### 2.3. LPM Modelling and Effect Detection

An LPM model is built in two parts: control group behaviour, C=c, and additional behaviour seen in treatment groups, C=t. Combined, these give the number of voxels associated with treated or untreated behaviour for each ADC value, *A*, and visit, *V*, each tumor then has a histogram that can be described as:H(A,V)≈M(A,V|C=c)+M(A,V|C=t)
$$M\left(A,V|C=c\right)=\sum_{c}P\left(A,V|C=c\right)Q_{c}$$
$$M\left(A,V|C=t\right)=\sum_{t}P\left(A,V|C=t\right)Q_{t}$$

M(A,V|C=c) and M(A,V|C=t) are built using control and treatment data respectively. Treated tumors may exhibit some behaviour or contain tissue that is unaffected by treatment. The second stage of training therefore fits both the control and treatment parts to describe tumors in the treatment cohort. In this way, treatment-specific components only describe additional behaviours not seen in M(A,V|C=c). The total number of voxels exhibiting treatment is $Q_{T}=\sum_{t}Q_{t}$.

Parameters are estimated via a Likelihood-based algorithm (Expectation Maximisation) and a Leave-One-Out (LOO) generalisation (Figure 2). It determines how many model components are required, the shape of probability mass functions and the weighting quantities that provide good descriptions of the data. A $\chi^{2}$ per d.f. goodness-of-fit is used. A sufficient model is selected from a large number of possible solutions from multiple random restarts to avoid poor local minima. Linear models are commonly degenerate with multiple models capable of describing the same data. We seek those solutions that are the most generalisable to LOO examples. A model is deemed sufficient if it: has a low fit value; does not produce LOO outliers; and has a Poisson residual distribution (checked via Bland-Altman analysis). If quality control measures are satisfied, the uncertainty, $Q_{T}\pm \sigma$, is computed using error propagation.

ADC changes due to treatment are assessed in three ways: (1) LPM estimates of $Q_{T}$ provide a lower-bound on change on a per-tumor basis. These can be divided by the error to give a Z score (standard deviations of change away from zero); (2) means, 95th percentiles and volumes are computed from whole tumor ADC distributions, H(A,V); (3) means, 95th percentiles and volumes are computed using the LPM decomposed sub-distributions, M(A,V|C=c) and M(A,V|C=t). Differences between treatments and control parameters are then assessed via *t*-tests.

### 2.4. Monte Carlo Simulation

Any comprehensive testing of the algorithm’s performance is limited by the finite quantity of available data and the quality of ground truth. To test more thoroughly, we simulate further data with known ground truth using Monte Carlo methods. LPM models can be used to simulate further histograms. Real control tumors and strongly responding RT treated tumors (Z>3) from the V1–V3 model are used as a basis. To create a simulated distribution, pairs of tumors are randomly selected. Their LPM mixing weights (*Q*) are interpolated or extrapolated randomly so that they deviate by no more than half of their original value. These coefficients mix corresponding PMFs, with individual histogram bins then synthesised using a Poisson random number generator. Control cohorts are made using only control tumors. Treatment cohorts are made using *n* RT pairs and N−n control pairs, simulating responders and non-responders. 10 cohorts are generated per set of test parameters.

ADC distributions for cohorts of different sizes, *N*, and number of responding tumors, n∈N, were created. Investigation of responding subsets, n∈N, covered a range of 5 in 12, up to 12 in 12. Cohort size tests covered the range N=2 to N=12. We were free to select multiple *p*-value thresholds, with lower thresholds corresponding to more stringent evidence with lower false positive rates (FPR). We apply thresholds of 0.05, 0.01 and 0.001 giving FPR 1 in 20, 1 in 100 and 1 in 1000.

### 2.5. Power Calculations

We can determine the minimum data required to reach desired levels of statistical significance with two power calculations. Firstly, via the LPM error theory described in [11], the power (in terms of Z score) as a function of independent voxels present, $Q_{total}$, is approximately
(1)$$Z=\sqrt{Q_{total}\left(T_{*}-\frac{{\left(1-T_{*}-C_{*}\right)}^{2}}{4C_{*}}\right)}$$
$$Q_{total}=Q_{T}+Q_{C}$$
$$T_{*}=<P\left(C=t|A,V\right)P\left(C=t|A,V\right)>$$
$$C_{*}=<P\left(C=c|A,V\right)P\left(C=c|A,V\right)>$$

The dominant variable that predicts the power of a *t*-test is the sample size (number of tumors). LPM, however, operates on a per-tumor basis and its power is independent of the size of cohorts. The dominant LPM variable determining the power to quantify tumor change is the number of independently sampled voxels in a tumor. Here, as the total quantity of voxels goes up, the attainable Z-scores also rise with the square-root of the quantity. The $T_{*}$ and $C_{*}$ are linked to the ambiguity between control and treated distributions. If control distributions look very similar to treated distributions then the Z-scores are penalised. However, in the case of completely unambiguous data, where the distributions have no overlap, the Z-scores grow perfectly as the square of the voxel count, following from Poisson behaviour. In order to use this calculation, an estimate of the ambiguity terms can be taken from similar past experiments.

Secondly, if only a subset of tumors respond, how large a cohort is required for at least one detection is given by Binomial theory. The probability of observing at least 1 response in a cohort of *N* is
(2)$$P_{detection}=\sum_{r=1}^{N}F^{r}{\left(1-F\right)}^{N-r}\frac{N!}{(N-r)!}$$
where $P_{detection}$ is the probability that at least 1 response will be measured and *F* is the fraction of the cohort expected to respond.

## 3. Results

### 3.1. LPM Determines Multi-Time Point Model Complexity

Model selection curves can be seen in Figure 3. For V1–V2 and V1–V3 models, the control parts have the same level of complexity and required 6 components to create a sufficient description of the data. Regarding the treatment part, the use of Visit 3 data in both V1–V3 and V1–V2–V3 models increases the required model order, suggesting that there is more information (variability) within the third scan than there is in the second. The most complex model was the V1–V2–V3 version, requiring 9 components for the control part and 13 for the complete model.

### 3.2. Confining Parameter Analysis to Treated Tissue Improves t-Tests

We assessed if LPM could improve estimation of response to therapy using the summary parameters typically used in conventional analysis of ADC images. LPM decomposed Calu6 distributions into parts indistinguishable from controls (considered ‘unaffected’) and parts different from controls (considered ‘affected’), as seen in Figure 4. Mean ADC, 95th percentile and tumor volume was computed for whole tumors, and for unaffected and affected parts separately (Figure 5). A conventional analysis was then performed on these extracted parameters.

For volume, full distributions showed that ATV treated tumors had growth rates indistinguishable from controls, RT had some growth inhibition and FCRT growth inhibition was modest. Growth was observed in all cohorts of treated tumors in their unaffected components. Considering affected tissue, RT tumors showed greatest size reduction. The main benefit of separating into habitats was seen when comparing ADC data. Modest increases in mean ADC were observed with RT and FCRT but not ATV versus control in full distribution data. However, when affected data only were compared the increase was greater in RT and FCRT and in addition was observed in ATV treated tumors as well. Furthermore, the habitats affected in ATV treated tumors were significantly higher on average than those affected in RT and FCRT. ADC values in unaffected habitats were indistinguishable from control behaviour.

Figure 5B shows that *p* values achieved via *t*-tests, comparing control and treated tumors, are more significant when parameters are confined to treated habitats, rather than the whole tumor. This effect is more marked when comparing V1–V3 compared to V1–V2. No substantial difference was noted for tumor volumes. These data demonstrate that parameters used in conventional testing have increased statistical power when analysis is confined to treated habitats, as defined by LPM.

### 3.3. LPM Detects Biological Response Rates across a Range of Therapies

We evaluated if LPM could distinguish biological response rates across different therapies all examined in the Calu6 model. Figure 6 shows ‘affected’ volumes consistent with biological response for control tumors and those receiving RT, FCRT and ATV. Data are presented for V1–V2, V1–V3 and V1–V2–V3 scan combinations. All control data had ‘treated’ volumes consistent with noise limits.

Biological responses were observed in all therapeutic groups and LPM revealed differences between them in a way not possible to detect with conventional cohort-based analysis. Response differed by volumetric extent, temporal onset/duration and by overall response rate. RT and FCRT cohorts behaved in similar ways by day 7 (V3) with 100% of tumors showing some effect. However, only 3 in 9 RT tumors showed effects at day 3 (V2) compared to 5 in 6 FCRT tumors. FRCT had earlier effects than RT, with corresponding increases in Z scores, often above 10. Most tumors had between 30–80% of tumor tissue affected, an observation that cannot be demonstrated using conventional summary parameters.

The V1–V3 and V1–V2–V3 models for RT and FCRT had similar responses, with one RT tumor changing from a non-responder to a responder. The addition of three time points (additional data) thus slightly increased the power to discriminate affected from unaffected tumors. The ability to incorporate additional time points in this way is a strength of the LPM approach that can not easily be achieved using the inherently pair-wise *t*-test.

For ATV, significant effects were seen in around 70% of tumors but the majority had treatment effects in 20% of the tissue or less. Only two tumors reached high significance, exceeding Z scores of 10. The inclusion of V3 data appears to have negligible affect on response detection, suggesting that ATV acts quickly (when it does act) and most of the information regarding change is already present by V2. For comparison, *p* values are listed alongside effect size and Z scores in Appendix A.

### 3.4. LPM Detects Biological Response Rates across a Range of Tumor Models

We evaluated if LPM could distinguish biological response rates across a single therapy in five different xenograft tumor models. We chose a single fraction (10 Gy) of RT because this is a well-understood treatment that is known to consistently induce a rapid treatment effect in many preclinical models [17]. We examined two syngenic tumor models (CT26 and 4T1) and two further ATCC human cell xenograft models (U87 and HCT116) in addition to the Calu6 cohorts, to span 5 varied tumor models.

Figure 7 shows ‘affected’ volumes consistent with biological response for control tumors and those receiving RT for V1–V2 only. Z scores and *p*-values for each tumor are listed in the suplimentary material. Control data had ‘treated’ volumes consistent with noise limits for 71/72 tumors across the five cohorts of control tumors. Just 1/72 individual tumor had Z score >3.0 which was CT26 with Z score of 4.34.

Cohort *t*-test evaluation showed that RT increased mean ADC significantly in three models (U87 *p* = 0.0121; HCT116 *p* = 0.0002; CT26 *p* = 0.0006) but only showed a trend with Calu6 (*p* = 0.0545) and 4T1 (*p* = 0.1003). LPM showed evidence in response and the per tumor level that was not detected by cohort-based analysis in the Calu6 and 4T1 models, as well as in the other 3 models. LPM revealed gradation of biological responses with Z scores >3.0 seen at day 3 in 50% (11/22) of CT26 tumors, 30% (3/10) of 4T1 tumors, 75% (9/12) of U87 tumors and 93% (14/15) of HCT116 tumors, compared with the 33% (3/9) of Z scores >3.0 in Calu6 tumors. Overall, typically 20–80% of the tumor tissue was affected when Z scores of 3.0 were recorded. This is information that has previously not been possible to derive from conventional MRI methods such as diffusion weighted imaging. For comparison, *p* values are listed alongside effect size and Z scores in Appendix A

### 3.5. LPM Achieves Consistently High True Positive Rates

Monte Carlo data reveal true-positive rates (TPR) of change detection. Two factors were investigated that may impact the ability to detect biological response: heterogeneity in treatment response, where only a subset (n∈N) of a cohort show signs of therapy-induced change; and the effects of changing the total cohort size (*N*).

Figure 8A shows TPR as a function of n for common significance levels. We considered scenarios with cohort sizes N=12 and varied the subset of tumors that had biological response, *n*, starting with less than half responding. We calculated TPR for *t*-test changes in volume, mean, 95th percentile and LPM responding volumes on tumor and cohort level. At the 0.05 level, change in volume showed the worst TPR, with up to 20% detection when n/N was 9/12 or less. LPM on individual tumors outperformed change in 95th percentile and was comparable to mean changes *across the entire cohort*. LPM across cohort acheived very high TPR, detecting 10 out of 10 simulated cohorts. As the threshold increased TPR declined substantially for *t*-tests, whereas the LPM for individual tumors had minimal reduction in TPR and across cohort retained a very high TPR. Figure 8B shows the area under the TPR curve (AUC) as a function of *p* value thresholds. LPM TPRs were relatively independent of the proportion of biological responders in a cohort.

Figure 8C show TPR as a function of *N*, with all tumors being biological responders. At the 0.05 level, change in volume showed the worst TPR again, but with little to distinguish cohort level *t*-tests from LPM until the cohort size was *N* = 4. Figure 8D shows the AUC as a function of *p* value thresholds. TPR declined substantially for *t*-tests for stricter thresholds, whereas the LPM did not. LPM results were relatively independent of cohort size.

LPM detection of change based on individual tumors consistently and significantly outperforms cohort level *t*-test changes in volume, mean and 95th percentile across a range of statistical thresholds. In particular the differences at 0.01 and 0.001 are important as these levels of significance substantially reduce the risk of false positive responders. The per-tumor approach gives LPM the ability to detect tumor change independently of treatment cohort size and biological response rate, in contrast to *t*-tests using simple distribution parameters on cohorts.

### 3.6. Validation of Power Calculations

The power calculation of Equation (Equation 1) was applied to a range of tumor sizes (Figure 9A) by estimating $C^{*}$ and $T^{*}$ from the cohorts used in this study.

The statistical power attainable for different voxel counts using LPM on individual tumors is shown in Figure 9A. The curves show the Z-score predictions as a function of total voxel count. The points plotted on the curves show the location of the real study data, confirming that the predictions are consistent with the power achieved in practice. Some treatments generate large biological response effects and accompanying large Z scores, requiring as few as 500 voxels (assuming voxel each is independent) to reach Z scores above 6. This is less than half the data used in previous studies [13]. As a consequence, it may be possible to reduce the time between scans and the size of tumors at the initial scan or increase voxel size, which may enable LPM to be applied to data with larger voxels such as seen in positron emission tomograpy. This possible reduction in study length does, however, assume that the ambiguity terms, $T^{*}$ and $C^{*}$, remain approximately the same for smaller tumors. Figure 8 part A shows that higher ambiguity (lower values of $T^{*}$ and $C^{*}$) associated with the different tumor models significantly reduces the power of LPM analysis. Reducing ambiguity may be achieved by selecting visits which emphasise differences between control and treatment distributions.

The individual tumor power predictions cannot alone specify the power of an experiment. If only a subset of tumors exhibit a biological response then we still require a cohort large enough to ensure that some responses will be seen. The cohort sizes needed to identify at least one responder based upon power calculation Equation (Equation 2) is shown in Figure 9B. From this we see that conventional cohort sizes (*n* = 10) are still required to detect at least 1 responding tumor with 90% confidence if only 20% of tumors are likely to show biological response. However, such responses would be highly diluted by non-responders in a conventional *t*-test, so the LPM analysis would still be beneficial. If more than half of tumors are expected to respond then treatment cohorts as small as five may be used with over 95% confidence that at least one tumor will show significant signs of biological response.

### 3.7. LPM Makes Small N Co-Clinical Trials Feasible

LPM has sufficient power to detect effects in samples of less than four tumors, as seen in simulated data and predicted by power calculations. Current deployment of preclinical imaging tends to require groups of 8–12 per cohort, making evaluation of multiple treatment arms time consuming, expensive and require substantial numbers of mice. We therefore investigated the utility of LPM as an analysis technique in co-clinical trials.

Prospective data was acquired in further cohorts of Calu6 tumors for control mice (*N* = 7) and three treatment groups, to form a mock co-clinical trial. We evaluated the prospective control data and compared it to the original control cohort (*N* = 15). The fit of three prospective control tumors was comparable to the original cohort controls; these were considered ‘strongly representative’ of the original control cohort data. A further four control tumors had fits of 1.9 to 2.4 that were outside the original cohort; these were considered ‘less representative’ (Figure 10A). Responding volumes were computed for all 7 prospective controls and were shown to be compatible with original controls (Figure 10B).

We evaluated cohorts of three new therapeutic groups with three tumors per cohort, guided by the above experiments of TPR and power calculations. LPM detected 3/3 biological responders in the atovaquone and radiotherapy treated group (ATV-RT), 2/3 biological responders in the AQ4N 60 mg/kg single dose group and 3/3 biological responders in the banoxantrone 20 mg/kg daily group, when the strongly representative controls were used (Figure 10C). We then ran an equivalent experiment using the less representative controls. The biological response rates were unchanged for the ATV-RT group and the banaoxantrone group treated with 60 mg/kg. However the biological response rate for the banoxantrone group was 1/3 rather than 3/3 (also Figure 10C). This shows that LPM could detect biological response in cohorts of *N* = 3, for new therapies (AQ4N), different dosing regimens (single dose of 60 mg/kg vs. 3 doses of 20 mg/kg daily) and a new combination of therapeis (ATV with concurrent RT). Furthermore, equivalent data were possible with sub-optimal controls (poor matches) but that LPM was liable to misclassifying when the overlap between control and treated tumor distributions was greater. Z scores were computed (Figure 10D) and comparison of these scores between strongly and less representative controls showed that the representative control group consistently detected greater responding volumes than the less representative group.

## 4. Discussion

Preclinical imaging is a powerful tool for investigating changes in tumor biology induced by RT and drugs [18]. Ethical and practical constraints limit the numbers of animals available in research so experiments typically are limited to cohorts of 8–12 per group [19,20]. Tumor heterogeneity and rapid physiological changes can limit the ability of cohort *t*-tests to detect effects using such limited data.

### 4.1. LPM Is an Alternative Paradigm for Pre-Clinical Cancer Research

Medical research has a long history of applied statistical methods and relies upon a well-established base of techniques. Many of these techniques were devised in an age of ‘small data’ and before modern computing. Amongst these is the *t*-test, which allows cohorts of simple Gaussian distributed values to be compared to one another. Importantly, a *t*-test facilitates the estimation of the statistical significance of any differences between cohorts. The ability to produce *p*-values is an essential property of traditional methods and one reason why they are widely accepted as valid scientific tools. In contrast, modern machine learning and AI has been developed in an age of ‘big data’ and has been applied to problems that are far more complex than simple value comparisons. These modern methods can adapt and learn to describe non-Gaussian behaviour. However, less attention has been paid to the estimation of uncertainty and therefore it is difficult to compute *p*-values from current machine learning systems. This is a significant disadvantage in a quantitative scientific setting, especially if limited by ‘small data’, as is often found in medical research.

Linear Poisson Modelling is based upon traditional statistical theories: regression, error propagation and hypothesis testing [11]. As such, it provides essential outputs for science, including *p*-values. But it is also a learning system that can adapt to non-Gaussian behaviour. In contrast to Deep Learning, LPM also works with very small datasets. These properties make LPM an ideal tool for the type of pre-clinical research presented here. LPM is an adaptive learning system and also an efficient alternative to *t*-testing. Simple parameters, such as means and percentiles, are readily computable and have intuitive meanings. However, *t*-test results using them are orders of magnitude less significant than LPM results for the same images. LPM’s scientific credentials and learning capabilities provide a new paradigm for pre-clinical imaging research. At the time of writing, we are unaware of any other machine learning or AI system capable of achieving the results we have presented here.

### 4.2. Origins of Additional Power of LPM

A *t*-test result may be significant for two reasons: (1) there is a consistent change in most or all tumors away from control behaviour; or (2) there are very large changes in a few tumors, large enough to compensate for a lack of change in others. The summary outputs of a *t*-test do not allow a researcher to tell the difference between these two scenarios. In contrast, an LPM analysis does enable this distinction, as non-responding tumors do not affect the power to identify individuals that are responding.

Statistical significance is a function of signal-to-noise. Given that we are using the same source data in both *t*-test and LPM analyses, the large increases in statistical power when using LPM may invite scepticism. The signal *t*-tests measure is the mean difference between two populations with respect to the variance within those populations under a strong Gaussian assumption. LPM models non-Gaussian behaviour allowing it to utilise greater amounts of information. The modelling of ‘affected’ and ‘unaffected’ behaviours attributes signal variation more effectively. This effectively increases the signal-to-noise when using LPM, as variations that LPM models as being valid signal is confused with noise when using a *t*-test.

Gaussian data are fully described using simple parameters: mean, measure of spread and normalisation. However, histograms from ADC maps are not Gaussians. Condensing such data into means and spreads necessarily discards useful information. LPM retains information by creating bin-by-bin generative histogram models. Non-Gaussian variations that increase spread are interpreted by *t*-tests as noise and reduce statistical power. This is true even when variations are allowable parts of signal behaviour, such as biological differences in natural history growth of tumors. LPM learns non-Gaussian variations in control behaviour and can subsequently remove them from treatment cohorts. The remaining Poisson perturbations around LPM fits are also predictable. As allowable signal variations have already been modelled, these Poisson sources of noise are far less than the spread estimated when using a *t*-test.

### 4.3. New Experimental Designs

A number of factors can have an impact on future experimental designs: the ability to separate affected and unaffected parts of distributions; the ability to combine and fit different numbers of time points; and the cohort size independence of the power when using LPM. In this new regime, even subtle therapeutic effects are likely to be detected so long as at least some affected tumors are present. We may therefore speculate on possible uses.

Building multi-time point models of data that show very small effects may provide methods to investigate new behaviours previously masked by ambiguity. Parameters such as means may show clearer trends, or even trends running counter to conventional analyses due to the removal of unaffected tissue. Studies may be designed to apply a range of statistical analyses to affected only tissue.

LPM’s additional power and the ability to identify individual biologically responding tumors provides opportunities for new study designs. Conventional cohort sizes can be used to create control models, but this data suggests that small treatment groups in the order of *N* = 3 can be considered. This may provide the basis for running co-clinical trials. In cases where only a subset of a cohort show measurable changes, LPM can still be used in larger cohorts to place tumors in order of response rate.

### 4.4. 3Rs and Cost Benefits

A key benefit of LPM is the potential to reduce and refine the use of animals (‘3Rs’ [14]). Since parameters such as ADC, *K*${}^{trans}$ and T1 show considerable overlap between control and treated tumors, cohort sizes tend to be around one order of magnitude or above [17]. LPM provides a solution for sensitive tests with small *N* and enables calculation of parameters such as biological response rate, hitherto not possible with many functional imaging biomarkers that exist on a continuous scale. This facilitates reduction and refinement in multiple ways. The option to simply use fewer animals is clear. There is also the option of using the same number of animals as in conventional trials, but testing a wider range of treatments using them. For example a traditional study might use 36 animals in three groups of control (*N* = 12), treatment 1 (*N* = 12) and treatment 2 (*N* = 12), whereas LPM would enable a control model group (*N* = 12) to then be compared with approximately 6–8 different treatment groups (each *N* = 3–4) and provide evidence of significant treatment efficacy in many more therapies despite still using the same overall number of animals (here, *N* = 36).

## 5. Conclusions

We have applied LPM analysis of volumetric and ADC MRI data of Calu6 tumors to evaluate multiple therapeutics, and have compared the effects of RT in multiple tumor models. We have shown that LPM: (1) can differentiate between control growth and the effects of treatment, adjusting model complexity to best describe training data; (2) provides a method to confine estimation of parameters (volume, mean, 95th percentile) only to tumor habitats that exhibit treatment effects; (3) can increase the amount of information extracted from images (in comparison to *t*-tests) to increase statistical power; (4) can use concatenations of multiple time points, rather than being limited to the 2 time point used by *t*-tests, with resultant improvement in the sensitivity to detect change; and (5) facilitates small N experiments, where treatment cohorts as small as *N* = 3 can be analyzed with high confidence. Limitations of this technique are that at present the technique does not label individual voxels with high enough levels of confidence to assign treatment effect or no treatment effect to an individual voxel and thus produce a spatial map of change following treatment. If further refinement of the technique does enable spatial mapping then validation with histopathology will be required. Finally, as with all biomarkers, assessment of measurement repeatability will need to be performed as part of the translation process.

The net effect of these benefits is that LPM provides a platform to enable prospective co-clinical trials [15] that evaluate response to a range of investigative therapies. To be effective, larger ‘master’ control cohorts (using current typical mouse numbers) are examined along with multiple small treatment groups. This approach not only makes such studies practical and efficient but also has clear implications for the well-being and ethical treatment of animals by implementing two of the ‘3Rs’ principles of reduction and refinement.

## Figures and Tables

**Figure 1 cancers-14-02159-f001:**
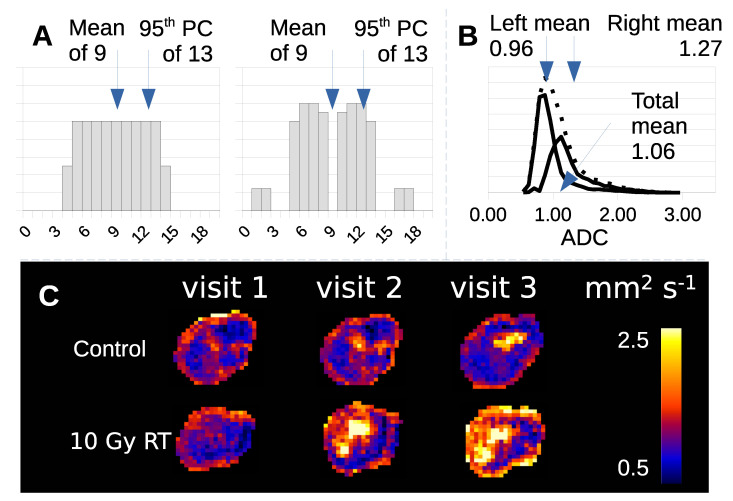
(**A**) Example of two data distributions (arbitrary units) that have the same mean, 95th percentile and integral, but are clearly different from one another. (**B**) Histogram of ADC data from a tumor treated with RT (dotted distribution) decomposed into treatment effect (left distribution) and no-effect components (right distribution). (**C**) ADC maps of control and RT treated Calu6 tumors have distinct habitats with different tumor microenvironments.

**Figure 2 cancers-14-02159-f002:**
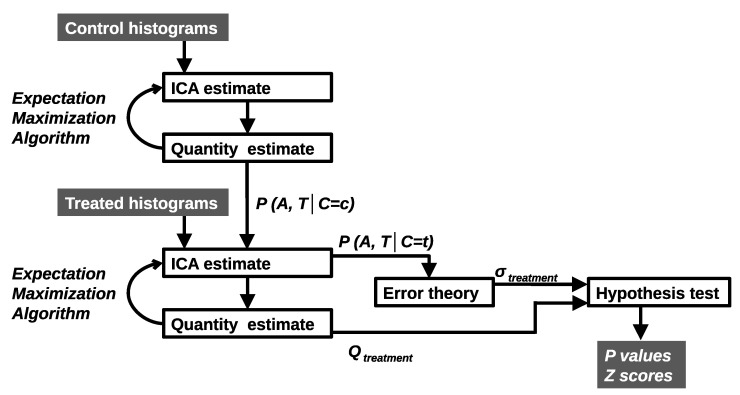
Control histograms are deconstructed giving components describing untreated behaviour. Histograms are added to create additional components to describe treated behaviour. All components are combined with Bayes Theorem to give per bin, per time point (A, T) probabilities of classification (C = c control or C = t treatment). The error theory is used to produce hypothesis tests for the significance of treatment effects.

**Figure 3 cancers-14-02159-f003:**
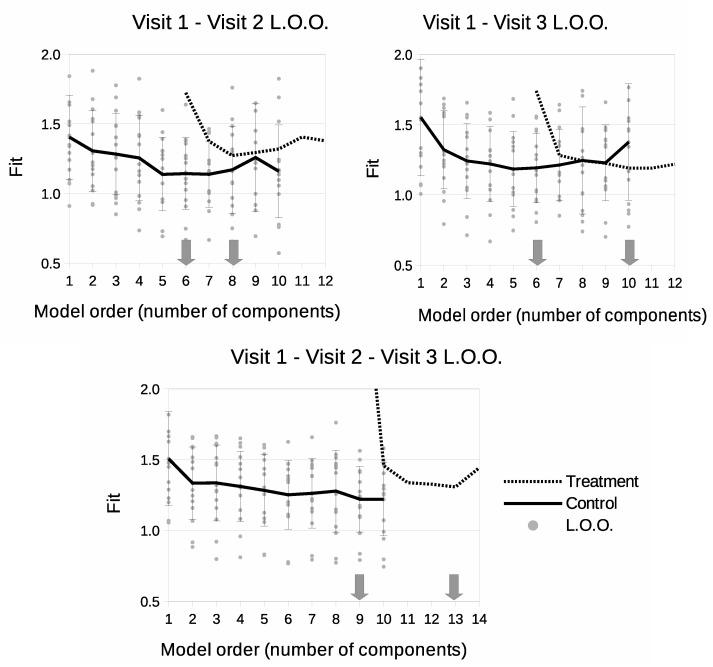
Leave-One-Out model selection summary showing goodness-of-fit (square-root of $\chi^{2}$ per degree of freedom) as a function of number of model components. Solid lines show model fits for control data only. Dotted lines show fits as treatment data is introduced to the model. Grey dots show individual fits for control LOO samples. ‘v’s indicate time points included in the model.

**Figure 4 cancers-14-02159-f004:**
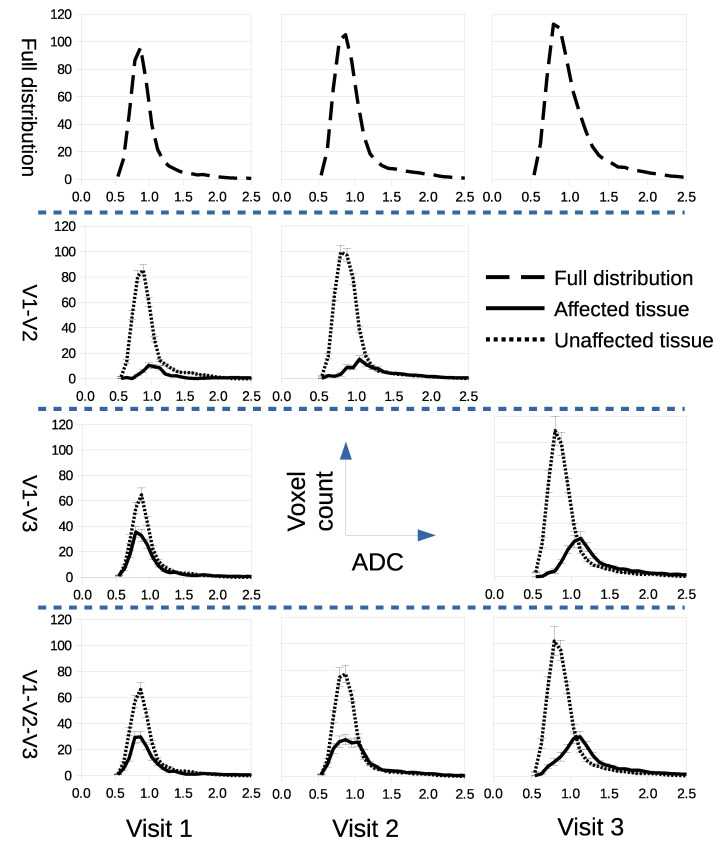
LPM decomposes ADC distributions into treatment-affected parts and parts that have no measurable differences from control distributions (unaffected). Rows from top to bottom show the full ADC distributions before decomposition, followed by alternative decomposition models for scan combinations V1–V2, V1–V3, and V1–V2–V3.

**Figure 5 cancers-14-02159-f005:**
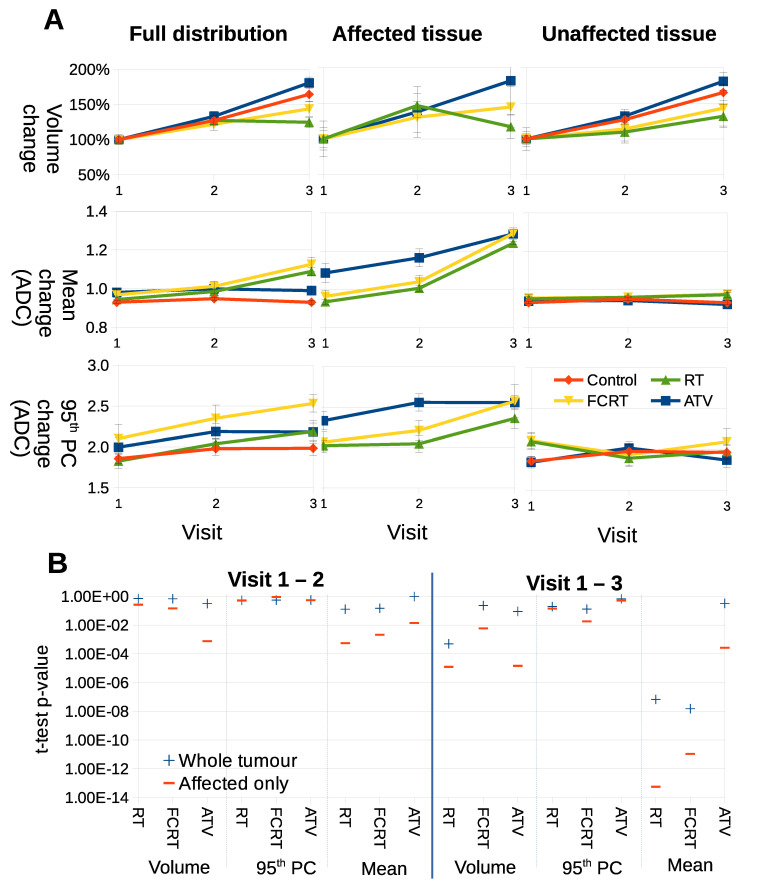
(**A**) Parameters computed using full ADC distributions or sub-distributions as decomposed via LPM. Each curve shows the average behaviour of a different parameter (volume, mean and 95th percentile), as computed from each respective cohort: control, RT, FCRT and ATV. (**B**) *p*-values achieved via *t*-tests comparing control to treatments using basic parameters between different time points. From left to right, the columns show parameters for the full distribution (a standard analysis of ADC), from affected regions, and for unaffected regions.

**Figure 6 cancers-14-02159-f006:**
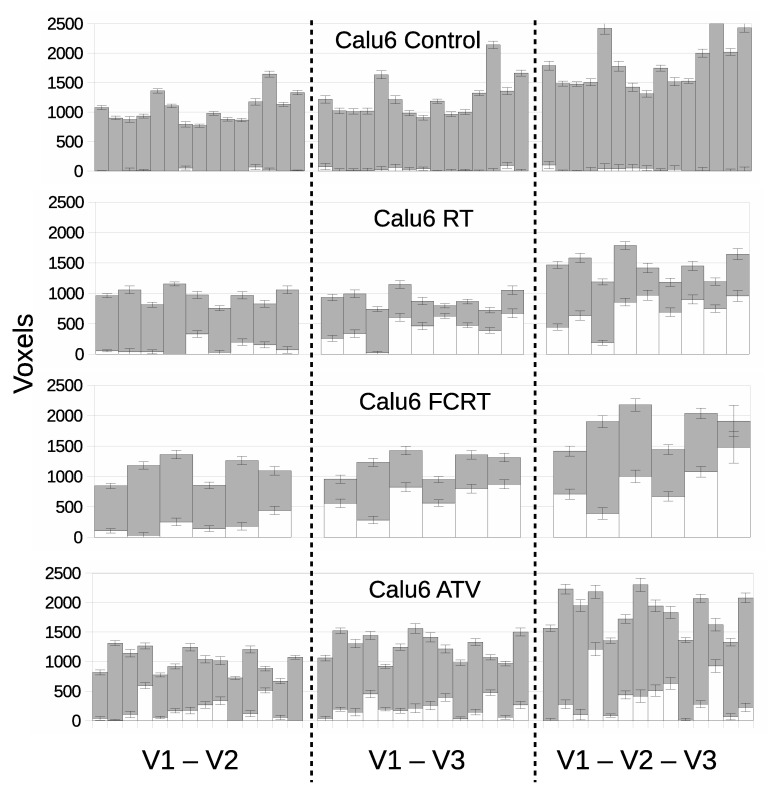
Per-tumor LPM assessment of treatment effects showing lower bound on affected tissue ($Q_{T}\pm \sigma$) in white and unaffected ($Q_{C}\pm \sigma$) in grey. Each bar represents a different tumor. The top plot confirms the null hypothesis of no treatment effects within the control cohort, followed by treatment cohorts showing responses. From left to right, cohorts are assessed using different pairs and triplet of visits corresponding to the V1–V2, V1–V3 and V1–V2–V3 models.

**Figure 7 cancers-14-02159-f007:**
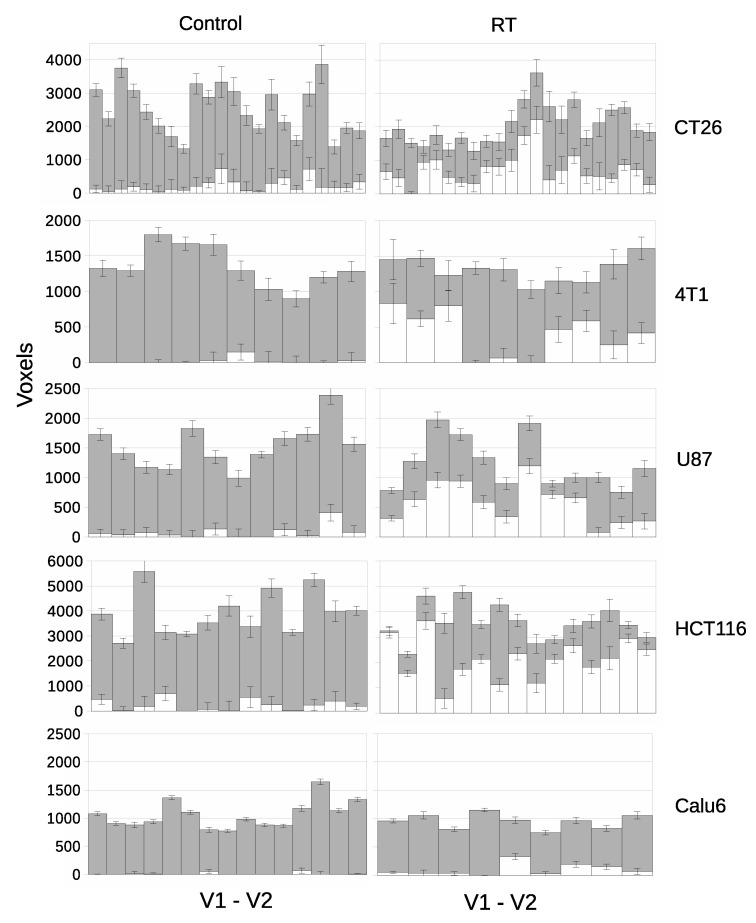
Per-tumor LPM assessment of treatment effects showing lower bound on affected tissue ($Q_{T}\pm \sigma$) in white and unaffected ($Q_{C}\pm \sigma$) in grey. Each bar represents a different tumor. The top plot confirms the null hypothesis of no treatment effects within the control cohort, followed by treatment cohorts showing responses. From left to right, control tumors and RT treated tumors. From top to bottom there are different tumor models.

**Figure 8 cancers-14-02159-f008:**
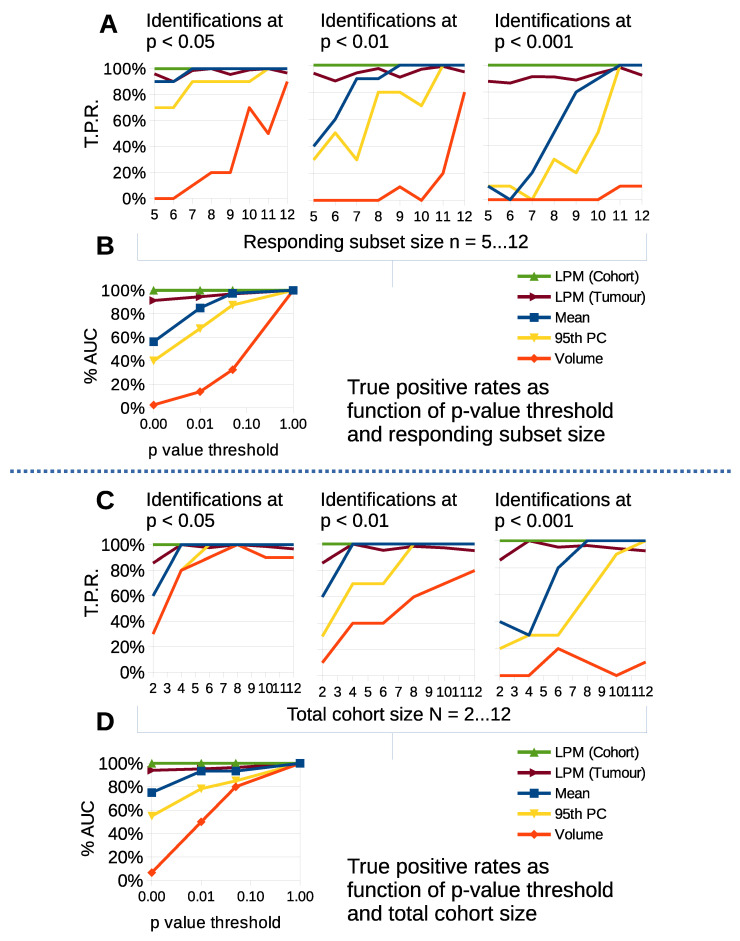
Simple parameter *t*-tests and LPM detections compared on simulated data as a function of responding subset size (n∈N), total cohort size (*N*) and statistical significance threshold for detection. Plots show true positive detection rates for null-hypothesis rejection. (**A**) TPR as function of responding subset size, *n*, down to less than half responding from a total of N=12. (**B**) Area under curve summary of TPRs as function of significance threshold found in (**A**). (**C**) TPR as function of total cohort size *N*. (**D**) Area under curve summary for (**C**).

**Figure 9 cancers-14-02159-f009:**
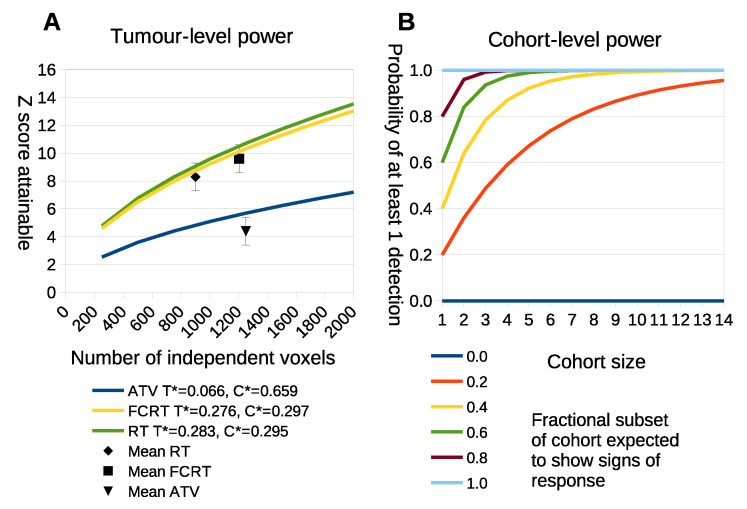
(**A**) The x-axis is the number of independently measured voxels within a tumor. The y-axis is the statistical power that is predicted to be achieved in standard deviations away from the null-hypothesis of there being no treatment effects. Each curve makes use of ambiguity terms computed from the three different treatment groups. The black points show where the real data lies, with error bars based upon the spread of Z scores seen within the respective cohorts. (**B**) The probability of observing at least one positive detection within a cohort when only a subset of tumors respond. The x-axis is the number of tumors within a cohort. The y-axis is the probability of at least one detection. Each curve represents a different level of fractional response at a cohort-level, i.e., the fraction of tumors that are likely to give a true response.

**Figure 10 cancers-14-02159-f010:**
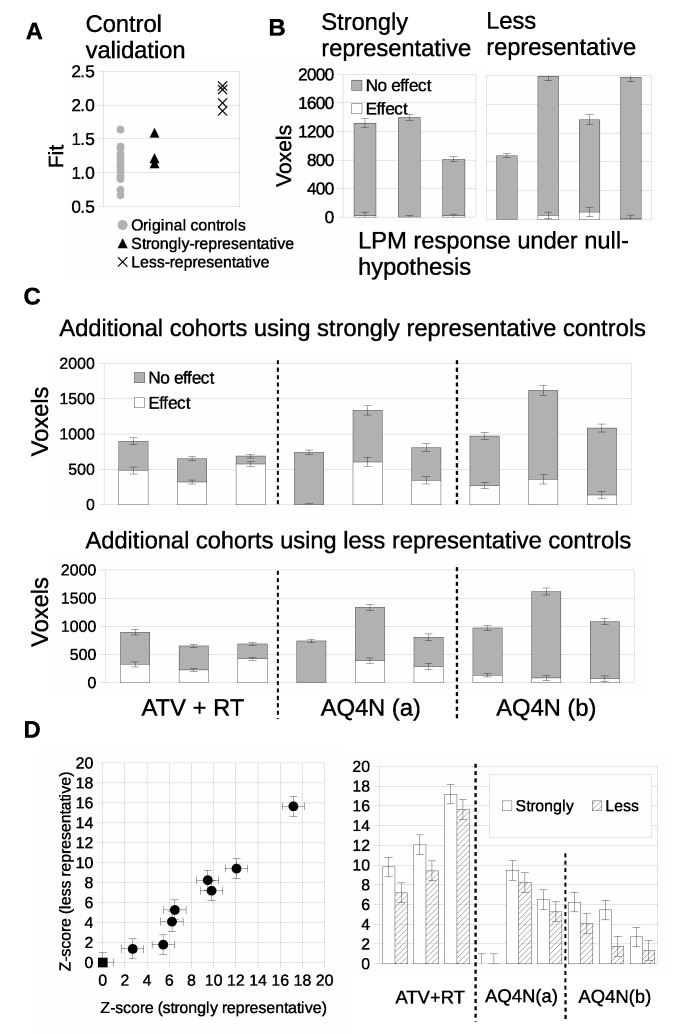
Validation and conformity checking of additional control tumors. (**A**) LPM goodness-of-fit ($\chi^{2}$ per d.f.) of additional control distributions compared against original controls (grey discs). The ‘strongly representative’ controls are the triangular points and the ‘less representative’ controls are the crosses. (**B**) Validation that when re-trained using additional controls, the LPM models show that both groups are consistent with there being no treatment effects. LPM analysis of additional treatment cohorts. (**C**) Volumetric assessment of tumor changes in response to the additional treatments with different control models (ATV with RT, AQ4N(a) with single dose of 60 mg/kg, AQ4N(b) 3 doses of 20 mg/kg daily). (**D**) Statistical significance, in terms of Z-scores, of treatment effects compared across both control groups models.

**Table 1 cancers-14-02159-t001:** Cell line specific MRI acquisition.

Xenograft Model	In Plane Maxtrix	Slice Number	In Plane Size	Slice Thickness	Volume
Calu6	64 × 64	15	0.5 mm × 0.5 mm	1.0 mm	0.25 mm${}^{3}$
U87	64 × 64	15	0.5 mm × 0.5 mm	1.0 mm	0.25 mm${}^{3}$
HCT116	64 × 64	7	0.4 mm × 0.4 mm	1.2 mm	0.192 mm${}^{3}$
CT26	64 × 64	10	0.5 mm × 0.5 mm	1.2 mm	0.3 mm${}^{3}$
4T1	64 × 64	10	0.5 mm × 0.5 mm	1.2 mm	0.3 mm${}^{3}$

## Data Availability

Data will be made available upon request to the corresponding author. LPM demonstration data is available at www.tina-vision.net.

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
