# Peer review of "Habitat Imaging of Tumors Enables High Confidence Sub-Regional Assessment of Response to Therapy"

_cancers, 2022, doi:10.3390/cancers14092159_

Round 1

Reviewer 1 Report

This paper describes a novel and inventive, resource-preserving method for deriving preclinical information. The paper is very well-written. It flows well and makes sense even to a clinician (non laboratory) scientist. I hope that this manuscript allows many scientists to do more research in a much less resource-intensive manner.

 My only suggestion is a small edit on line 84 (change "discernable" to "discernably") - otherwise it is perfect. 

Author Response

This paper describes a novel and inventive, resource-preserving method for deriving preclinical information. The paper is very well-written. It flows well and makes sense even to a clinician (non laboratory) scientist. I hope that this manuscript allows many scientists to do more research in a much less resource-intensive manner.

My only suggestion is a small edit on line 84 (change "discernable" to "discernably") - otherwise it is perfect

Thank you for the kind comments. We have made the edit as requested.

Reviewer 2 Report

In this paper, authors used LPM to identify responding volumes of tumors treated with RT in xenograft models of colorectal cancer.

However, the authors proved the subject matter with the same set-up and result presentation method in their previously published paper in Bioinformatics titled 'A new method for the high-precision assessment of tumor changes in response to treatment' (https://doi.org/10.1093/bioinformatics/bty115). Can authors describe the difference between the current work and their previous work in detail? In terms of novelty?

Author Response

REVIEWER 2

In this paper, authors used LPM to identify responding volumes of tumors treated with RT in xenograft models of colorectal cancer.

However, the authors proved the subject matter with the same set-up and result presentation method in their previously published paper in Bioinformatics titled 'A new method for the high-precision assessment of tumor changes in response to treatment' (https://doi.org/10.1093/bioinformatics/bty115). Can authors describe the difference between the current work and their previous work in detail? In terms of novelty?

Thank you for reviewing this manuscript. We are happy to clarify why the current manuscript is very different from the previous paper from Bioinformatics, which demonstrated technique feasibility and application to tumors treated with radiotherapy only in LoVo and HCT116 models.

The current new manuscript substantially extends this work, in particular:

1. We show that LPM enables extraction of simple distribution parameters (means, percentiles etc.) from only tumor tissue that is consistent with treatment effects. This is completely new.

2. We compare LPM across a range of therapies. This is completely new and provides evidence that LPM works with many treatments and tumour models.

3. We provide formal assessment of power calculations for LPM to justify small N experiments and plan future studies by providing estimates of how many and how large tumours need to be to achieve significant results. This is completely new.

4. We combine these different strands into a simulated co-clinical trial. This provides direct evidence that LPM can be used prospectively in studies with very small numbers of animals.

These key findings are described in the last section of the introduction and in section 5. Conclusions.

Reviewer 3 Report

General summary: In this effort the authors present a modeling technique to identify voxel-wise habitats corresponding to affected and un-affected tissue to various types of treatment and across numerous cell lines. The main finding is that through LPM they are able to (1) identify response rates to different treatments and (2) potentially reduce the number of animals needed for future studies. There are several experimental and mathematical strengths in this manuscript, however there are a few areas that should be addressed in revision.

Comment 1:  Figure 1-The caption and in-text description of this figure is insufficient to completely understand the different components. For panel (B) do the Left and Right mean correspond to a specific tissue type? Is the dashed line the total tumor histogram?

Comment 2: Figure 1, panel C.  Does the third column of images correspond to Visit 3?   Also the label for ADC units is cut off slightly in the manuscript.   In panel (B/C) it might be useful to indicate which voxels corresponded to the "left mean" and the "right mean" if they do indeed line up with images in panel C.

Comment 3: Line 42, what is meant by "Simple parameters"? This term is a little vague. Does this refer to voxel-wise ADC measures or statistical parameters (mean, standard deviation) describing the values of ADC within the tumor region of interest?

Comment 4: Line 120, what direction was the diffusion experiment collected in (and was the diffusion direction consistent between experiments)? Depending on the structure of the tumor tissue, the diffusion measurement might be effected by the orientation of the sample when a single direction is used.

Comment 5: Line 124-126. the formatting on the equation should be improved either using superscripted parameters or replacing "e" with "exp(-bD)". Additionally, equation parameters and variables should be italicized when mentioned in text (e.g., b, D, S, and S0).

Comment 6: Figure 6/7.  One thing that makes these figures a little hard to compare is that the number of voxels varies from tumor to tumor. Instead of reporting the number of voxels, it might be clearer to normalize each bar to the total number of voxels and report the fraction of the tumor that is affected and unaffected.

Comment 7: Additional discussion is needed to compare their proposed analysis techniques to other diffusion weighted analysis (e.g., functional diffusion mapping https://ascopubs.org/doi/10.1200/JCO.2007.15.2363 )  and habitat imaging approaches (ref. 17 Bruna et al, and  https://doi.org/10.3390/cancers14071837 )

Comment 8: Can you comment how your analysis may be affected by variability in ADC measurements from visit to visit?  Does the ADC in a healthy appearing tissue (say muscle) appear stable from visit to visit?

Comment 9: Once you perform your LPM analysis and decide the number of voxels in each group, is it possible to return back to the original ADC map and identify which voxels fall in the unaffected or affected group?

Comment 10: Have you done any histological validation of these tumor habitats?

Author Response

REVIEWER 3

General summary: In this effort the authors present a modeling technique to identify voxel-wise habitats corresponding to affected and un-affected tissue to various types of treatment and across numerous cell lines. The main finding is that through LPM they are able to (1) identify response rates to different treatments and (2) potentially reduce the number of animals needed for future studies. There are several experimental and mathematical strengths in this manuscript, however there are a few areas that should be addressed in revision.

Comment 1:  Figure 1-The caption and in-text description of this figure is insufficient to completely understand the different components. For panel (B) do the Left and Right mean correspond to a specific tissue type? Is the dashed line the total tumor histogram?

Thank you for pointing this out. We have amended the figure caption and in-text description as requested.

Comment 2: Figure 1, panel C.  Does the third column of images correspond to Visit 3?   Also the label for ADC units is cut off slightly in the manuscript.   In panel (B/C) it might be useful to indicate which voxels corresponded to the "left mean" and the "right mean" if they do indeed line up with images in panel C.

Thank you for pointing this out. We have amended the figure labels as requested.

It is not possible with current methods to show direct correspondence of the voxels in the density functions with the images (see our response to comment 9).

Comment 3: Line 42, what is meant by "Simple parameters"? This term is a little vague. Does this refer to voxel-wise ADC measures or statistical parameters (mean, standard deviation) describing the values of ADC within the tumor region of interest?

Thank you for pointing this out. We have clarified the text as requested.

Comment 4: Line 120, what direction was the diffusion experiment collected in (and was the diffusion direction consistent between experiments)? Depending on the structure of the tumor tissue, the diffusion measurement might be effected by the orientation of the sample when a single direction is used.

Diffusion gradients were applied in the read direction. This was consistent across all experiments. Since tumor tissue tends to have relatively low diffusion anisotropy the diffusion direction should have little effect on the value or interpretation of ADC values.

Comment 5: Line 124-126. the formatting on the equation should be improved either using superscripted parameters or replacing "e" with "exp(-bD)". Additionally, equation parameters and variables should be italicized when mentioned in text (e.g., b, D, S, and S0).

Thank you for pointing this out. We have amended the equation as requested. We have also italicized terms as requested.

Comment 6: Figure 6/7.  One thing that makes these figures a little hard to compare is that the number of voxels varies from tumor to tumor. Instead of reporting the number of voxels, it might be clearer to normalize each bar to the total number of voxels and report the fraction of the tumor that is affected and unaffected.

The presentation of these results in voxel counts was carefully considered and left un-normalised deliberately.

The LPM method is designed to output quantities, and importantly, error bars on those quantities. Normalising these plots would change our error model, artificially enforcing a binomial rather than Poisson interpretation of the data. By presenting voxel counts, the error bars can be presented on both affected and unaffected habitat volumes. If normalised, information would be lost from these plots as only a single error bar could be given per tumour. Information about the distribution of tumour sizes would also be lost from the plot, whereas in this format the total size of each tumour is clear.

We recognise that some readers may prefer percentages and have therefore tabulated normalised results and presented them at the end of the paper. We feel that this is more than just a stylistic issue, and that un-normalised plots are more informative.

Comment 7: Additional discussion is needed to compare their proposed analysis techniques to other diffusion weighted analysis (e.g., functional diffusion mapping https://ascopubs.org/doi/10.1200/JCO.2007.15.2363 )  and habitat imaging approaches (ref. 17 Bruna et al, and  https://doi.org/10.3390/cancers14071837 )

These approaches are mathematically and theoretically different to LPM, but try to do something similar at headline level – detecting tumor subregions that have differing underlying biology and hence differing response to therapy and prognosis.

We have expanded the penultimate paragraph on p3 and also include these references as well as Berry 2008 MRM.

Comment 8: Can you comment how your analysis may be affected by variability in ADC measurements from visit to visit?  Does the ADC in a healthy appearing tissue (say muscle) appear stable from visit to visit?

This is an important point. We have not addressed repeatability here. We have referred to this as part of a limitations section in the Conclusion, saying

As with all biomarkers, assessment of measurement repeatability will need to be performed as part of the translation process’ at the end of the first paragraph in section 5. Conclusions

We have not investigated LPM in healthy tissue in this current work. We acquire coronal slices of data from the superior aspect of the tumour downwards and so have very little non-tumor tissue within the field of view, preventing robust analysis of muscle in these mice.

Comment 9: Once you perform your LPM analysis and decide the number of voxels in each group, is it possible to return back to the original ADC map and identify which voxels fall in the unaffected or affected group?

Thank you for raising this point. We have given a great deal of thought to the process of projecting LPM models back into the spatial domain to create maps of responding and non-responding areas.

Because there is some overlap between the treated and control distributions in the histogram domain, it is not possible at present to label voxels with high enough levels of confidence as ‘definitely treatment effect’ versus ‘definitely showing no signs of treatment effect’. As such, any maps would have too high a level of uncertainty and potentially be misleading.

This type of spatial mapping may be more feasible if ADC was used in conjunction with some other parameters (such as quantitative T1 or T2, Ktrans etc). However, this would lead to LPM having to model multidimensional histograms which would require far more data. We are considering how this may be feasible going forward while keeping the numbers of animals realistic, but this is likely a separate piece of work.

We have referred to this in the limitations section in the Conclusion saying:

Limitations of this technique are that at present the technique does not label individual voxels with high enough levels of confidence to assign treatment effect or no treatment effect to an individual voxel and thus produce a spatial map of change following treatment’.

Comment 10: Have you done any histological validation of these tumor habitats?

Not as yet. This is planned for future work, if spatial mapping of treatment change becomes possible with a high enough certainty. We have added a line in the new Conclusions paragraph to this effect, saying:

If further refinement of the technique does enable spatial mapping then validation with histopathology will be required’.

Round 2

Reviewer 2 Report

Well done.